# Challenges and coping strategies faced by female scientists—A multicentric cross sectional study

Farah Naaz Fathima[1], Phyllis Awor[2], Yi-Chun Yen[3], Nancy Angeline Gnanaselvam[1], Fathiah Zakham[4]*

1 Department of Community Health, St John's Medical College, Bangalore, India, 2 School of Public Health, Colleague of Health Sciences, Makerere University, Kampala, Uganda, 3 Department of Life Science, Tunghai University, Taichung, Taiwan, 4 Department of Laboratory Medicine, Faculty of Medicine and Health Sciences, Hodeidah University, Hodeidah, Yemen

* fathiah_zakham@yahoo.com

## Abstract

Women can play a pivotal role in the progress and sustainability of the world if they are empowered through education and employment opportunities in Science, technology, innovation and through changing the social stereotypes that restrain them in certain workplaces. In the literature, few recently published studies exist that document the challenges faced by female scientists in their workplaces. The purpose of this study was to understand the challenges and coping strategies faced by female scientists around the world today, in order to contribute to their improved performance. A multi-centre electronic cross-sectional survey across 55 countries was conducted to profile female scientists and to identify the challenges that they experience throughout their career as well as the coping mechanisms that they use to overcome the barriers. A total of 263 female scientists from different countries across the world participated in our study and most participants were from the South East Asian Region. Most female scientists in our study belong to the middle and junior level career category and earning around 1250 USD per month. Most of the scientists reported availability of maternity leave at their workplace but less than a third reported presence of a creche at work. Workplace sexual harassment was reported by 24% of the study population. Work related stress (71.5%) and work life imbalance (46%) are also major challenges faced by female scientists. Self-confidence, dedication and hard work are the most commonly adopted coping strategy. Flexible work timings, woman-friendly management policies, fair appraisal and mentorship appear to reduce the work-related stress and improve work-life balance among female scientists. In conclusion, female scientists face numerous challenges, which can greatly affect both their individual and career growth. Intrinsic (personal) and extrinsic (institutional) factors are important for improving female scientists' wellbeing and productivity.

**Data Availability Statement:** Yes - all data are fully available without restriction. All relevant data are within the manuscript and its Supporting Information files.

**Funding:** The authors received no specific funding for this work.

**Competing interests:** The authors have declared that no competing interests exist.

## Introduction

Women's empowerment is one of the strategic challenges for sustainable development. Many countries have achieved gender parity in primary education, but a significant gender gap exists at higher levels of education. While the number of women graduating from universities with higher degrees is increasing, women scientists are grossly underrepresented in Science, Technology, Engineering and Mathematics (STEM) [1]. Systemic barriers prevent women from pursuing research oriented careers and gender disparities exist in employment, academic promotions or senior ship, funding opportunities and publications [2]. Further, efforts and working hours that most women spend in teaching, mentoring, collaborations and other academic activities are more than men do with lower pay [3]. Worldwide, women are underrepresented in first and last authorship, and in multiple authorship, women represent less than one third of the authors in publications [2].

In developing countries, the gender disparity in STEM is a multifactorial issue that includes familial, social, cultural and institutional factors that cannot be ignored or overlooked. Ultimately, lower representation of female scientists in STEM fields translates into fewer female role models for girls and limited mentoring opportunities [4]. Studies on the number of females in top positions in academic institutions suggest that unintended and subconscious gender bias is common and can result in barriers for women to be promoted, credited for their achievements, nominated for leadership positions or viewed as leaders [5].

Despite the great advances made by women today, the available evidence depicts that women are still under-represented in academia. The gender imbalance is especially true among those who pursue higher education and advance in their research careers [6]. The results of a study done by Pohlhaus et al. on the differences between males and female scientists in application, success, and funding rates for the US National Institute of Health (NIH) extramural programs showed that the percentage of females who were awarded early career grants was more than males (57% versus 42%). However, when it came to independent investigator initiated research grants the percentage of women-awardees was significantly lower than that for male scientists (27% versus 72%) [7]. The factors contributing to these differences include work life issues where women are required to show exceptional productivity during their childbearing years in order to secure the next position (i.e. starting and succeeding in their own lab). This is a uniquely female issue.

A randomized double-blind study conducted by Moss Racusin et al studied gender preferences while hiring laboratory researchers. This study showed that given equally qualified male and female applicants, science faculty members would show preferential evaluation and treatment of the male applicants for a managerial post. When compared to female applicants, the male applicants were rated as significantly more competent and hirable, offered a higher starting salary and more career mentoring. Both female and male faculty were equally likely to exhibit bias against the female candidate [8].

In an effort to adapt themselves to their work environment, female workers and scientists may have to make compromises that may impact their self-respect and even health. Challenging workloads both at home and at work can affect women's mental and physical health and often due to inequity in domestic chores with partners, it reduces marital bliss. High workload also affects work commitment and promotions, impacting women's career growth [9]. The health of working women is more affected as compared to men due to the double burden of unpaid care-giving work and paid work. Added to this is the stress of reduced chance of finding and keeping a job as a woman ages [10]. Political marginalisation of female workers and the unpaid nature of domestic work, together with rigid gender norms and stereotypes, often

undermine women's physical and intellectual capabilities, assisting the society in viewing her maternity and home-care as predominant functions [11].

While some previous studies have documented the challenges faced by female scientists [5–11], the purpose of this work is to profile female scientists working in STEM across different countries, to identify challenges faced by them, and to especially understand the coping strategies that they utilize along their career paths. This synthesis will help employers and female scientists to perform better.

## Methods

We conducted a multi-center electronic cross-sectional survey across countries to profile female scientists and to identify the challenges that they experience throughout their career and the coping mechanisms that they utilize. For the purpose of this study, a female scientist is defined as a female with expertise in one or more fields of STEM and currently employed in academia, government sector, industry or corporate sector. This scientist should have at least a master's degree in her specialty.

### Research ethics

Ethical approval for the conduct of study was obtained from the Institutional Ethics Committee at St John's Medical College, Bangalore [IEC335/2017]. All potential study participants were sent an electronic communication containing an invitation to participate in the study, a consent form, subject information sheet and the survey questionnaire.

### Sample size

A total of 606 female scientists in the field of science from among the personal contacts of the investigators were contacted. This was accompanied by personal emails followed by periodic reminders. Sample size estimation for the study was done with reference to a study of Pohlhaus et al [7] on the sex differences in funding for NIH Extramural Programs who reported a success rate of 22% among experienced female scientists in getting an NIH grant. Using 22% as an estimate of prevalence, and an absolute precision of 5% and an α of 5%, we estimated that the minimum sample size required for our study would be 263. We continued recruitment for our study until we received 263 complete responses from female scientists with a response rate of 43.4% (S1 Table).

### Survey design

The survey format was designed using Google forms. Variables studied included demographics of female scientists, workplace details, work-family conflict, gender discrimination, coping strategies and sexual harassment at the workplace. Data were extracted to Microsoft excel and analyzed. The sociodemographic profile of the female scientists, the challenges and the coping methods were described using descriptive statistics like proportions, means and standard deviation.

## Results

### Characteristics of female scientists

A total of 263 female scientists from 55 different countries across the world participated in our study. Majority of the study participants were from the South East Asian Region. The distribution of the study participants by region and county of origin is depicted in Table 1.

**Table 1. Distribution of the study participants by WHO region and nationality.**

| WHO region | Country | Number | Number by region (%) |
|---|---|---|---|
| African region | Algeria | 1 | 41 (15.59) |
| | Benin | 2 | |
| | Gambia | 1 | |
| | Ghana | 1 | |
| | Kenya | 7 | |
| | Mauritian | 1 | |
| | Mozambique | 1 | |
| | Namibia | 1 | |
| | Nigeria | 10 | |
| | Rwanda | 1 | |
| | South Africa | 1 | |
| | Swaziland | 1 | |
| | Tanzania | 1 | |
| | Uganda | 12 | |
| Region of the Americas | United States of America | 8 | 17(6.46) |
| | Argentina | 5 | |
| | Bolivia | 1 | |
| | Brazil | 1 | |
| | Canada | 1 | |
| | Chile | 1 | |
| South-East Asia Region | Bangladesh | 4 | 113(42.97) |
| | India | 104 | |
| | Indonesia | 4 | |
| | Nepal | 1 | |
| European Region | Austria | 1 | 28(10.65) |
| | Azerbaijan | 1 | |
| | Belgium | 1 | |
| | United Kingdom | 1 | |
| | Germany | 1 | |
| | Finland | 7 | |
| | France | 1 | |
| | Hungary | 1 | |
| | Italy | 5 | |
| | Portugal | 1 | |
| | Spain | 4 | |
| | Sweden | 1 | |
| | Switzerland | 3 | |
| Eastern Mediterranean Region | Egypt | 6 | 36(13.69) |
| | Iran | 1 | |
| | Jordan | 4 | |
| | Kuwait | 1 | |
| | Morocco | 4 | |
| | Pakistan | 5 | |
| | Sudan | 3 | |
| | Tunisia | 2 | |
| | Yemen | 10 | |

(*Continued*)

**Table 1.** (Continued)

| WHO region | Country | Number | Number by region (%) |
|---|---|---|---|
| Western Pacific Region | Australia | 1 | 28 (10.65) |
| | China | 1 | |
| | Hong Kong | 1 | |
| | Malaysia | 12 | |
| | Mongolia | 3 | |
| | Philippines | 3 | |
| | Taiwan | 6 | |
| | Vietnam | 1 | |
| Total | | 263 | |

The median age of the female scientists who participated in our study was 35.51 years [IQR: 30.39–42.37 years]. Around 40% of the study participants worked at a mid-level position in their institutions, with 34.2% at the senior level and the remaining (26.2%) at junior level. In addition to research work, 65.4% were involved in teaching, 26.6% in administration and 24.3% were involved in patient care responsibilities. The total number of years of experience ranged from 1 to 40 years with a median of 8 years [IQR: 4–14]. The reported median monthly income was 1250 USD [IQR: 800–2400 USD] and the mean number of hours of work per day was 8.4 ± 1.7 hours.

Around two thirds (64.3%) of the study participants were currently married, 29.6% were never married and 6.1% were divorced/separated. More than half of the study participants (54.8%) lived in nuclear families, 20.9% lived alone and the rest lived in nonnuclear families (24.3%). Close to a half of the female scientists (43.0%) did not have children, 21.7% had one child, 25.9% had two children and 9.4% had three or more children. Of the female scientists who had at-least one child (150), 39.9% of the cases had the youngest child in the under 5 age group, whereas 14.6% had children ≥18 years of age. The median age of the youngest child was 10 years [IQR: 2.13–18.75 years]

A majority of the participants (87.8%) reported that maternity leave could be available at the institutions where they work. However only 27.2% of the above could avail maternity leave for a duration of at-least 6 months. Around half (51.7%) of the female scientists reported that they had taken a break in their career for pregnancy and childcare responsibilities and the mean duration of the break was 2.3 ± 2.3 years. Around one third of the participants reported that their partner was also a scientist.

Around two thirds (65.3%) of the female scientists reported that they were helped in their careers by at-least one mentor. The most common mentor for the female scientists in our study was a senior male mentor (30.4%), followed by a senior female mentor (21.7%) and male and female senior mentors (13.3%). The female scientists in our study had a median of 8 research publications [IQR: 3–20] and had attended a median of 2 scientific conferences in the past one year [IQR: 1–3].

## Challenges faced by female scientists and coping strategies

We found that workplace factors played a major role in the life of female scientists as almost three quarters (71.5%) of them reported feeling work-related stress. Around half (48.3%) of the study participants felt that performance appraisal was done fairly among both sexes while only one third reported that the human resource policies in their organization were women friendly (34.6%) and that their management had a favorable approach towards careers of female

scientists (41.0%). One in four of the participants (24%) reported having experienced sexual harassment (verbal or physical) at their workplace and 32% reported being asked too many personal questions. Around half (56.3%) had flexible working hours and most (71.9%) regularly worked extra hours. Very few (10.6%) worked night shifts and few (28.9%) had a creche at their workplace. A majority (73.8%) reported that there was no disparity in pay between the two genders. Three quarters of the participants knew other women in science who have left their career due to some barriers. However, an overwhelming majority (87.5%) reported that they would recommend fellow women scientists to pursue career in science. The workplace factors among female scientists are depicted in Table 2.

Half of the female scientists in our study seemed to have a work life balance with 54% reporting that their families were able to adapt to their working hours and work demands and 44% reporting that they were able to spend productive time with their families. On the other hand, another half of the female scientists reported that their working hours prevent them from spending quality time with their family (46%) and that their work responsibility demands more of them than their responsibility with their families (49.8). Around one third (33.4%) were willing to trade their income for shorter hours at work in order to spend more time with their families. Some participants (40.3%) reported that taking care of their dependents had an impact on their work. Work-life balance among female scientists is shown in Table 3.

The common challenges faced by female scientists at work are lack of work-home balance, discrimination, harassment at workplace and a hostile work environment (Fig 1).

The common coping strategies faced by female scientists are attributes such as self-confidence, hard work and dedication, support for childcare and a supportive work environment (Fig 2).

Analysis of the perceived challenges by female scientists revealed that the top challenges were factors at the workplace. However, the main coping strategies were at the individual level and included self-confidence, hard work and dedication.

Overall, female scientists seemed to be fairly happy with the way their career has progressed (Fig 3). However, one in four female scientists reported that they were unhappy with their careers.

## Discussion

Women are globally under-represented in government, education and in the labour force [12]. We aimed to profile female scientists in STEM and their challenges as well as coping strategies.

**Table 2. Work place factors among female scientists.**

| Work place factors | n(%) N = 263 |
|---|---|
| Do you feel work related stress? | 188(71.5) |
| Is performance appraisal done fairly among both the sexes | 127(48.3) |
| Have you ever experienced sexual harassment (verbal, physical) at workplace | 63(24.0) |
| Is your working time flexible? | 148(56.3) |
| As a female scientist do you feel that you are being asked too many personal questions? | 84(32.0) |
| Do you have to put in extra hours before and after your standard working timings? | 189(71.9) |
| Do you have night shifts? | 28(10.6) |
| Is your management's perception towards females' career progression favourable? | 108(41.0) |
| Do you have a crèche at workplace? | 76(28.9) |
| Do you feel that HR policies are women friendly? | 91(34.6) |
| Is pay equal among male and female scientists? | 194(73.8) |
| Do you know of any women in science who have left their career due to some barrier? | 199(75.7) |
| Would you recommend fellow women in the future to pursue career in science? | 230(87.5) |

**Table 3. Work–Life balance among female scientists.**

| Work–life balance | n(%)<br>N = 263 |
|---|---|
| My working hours prevent me from having more quality time with my family. | 121(46.0) |
| My work responsibility time, demands more of me than my responsibility with my family | 130(49.8) |
| I would like to share the family responsibilities with my partner | 103(39.2) |
| My family is able to adapt to my working hours and work demands | 142(54.0) |
| I will trade my income for shorter hours at work to spend time with my family | 88(33.4) |
| I still spend productive time with my family even when I spend overtime hours at work or work over the weekend. | 118(44.8) |
| Taking care of my dependents affect my working time | 106(40.3) |

Our study is unique, in that it attempts to study women across different countries of the world, developed and developing. The investigators used google forms for data collection. This strategy is useful and feasible since most female researchers have access to the internet. They also have little for interviews due to their dual role of researcher and homemaker. Most of the study participants are from South East Asian region since the investigators belong to this region.

The top most coping strategies reported by female scientists in our study were at the individual level and included factors like self-motivation, confidence, assertion, faith, hard work

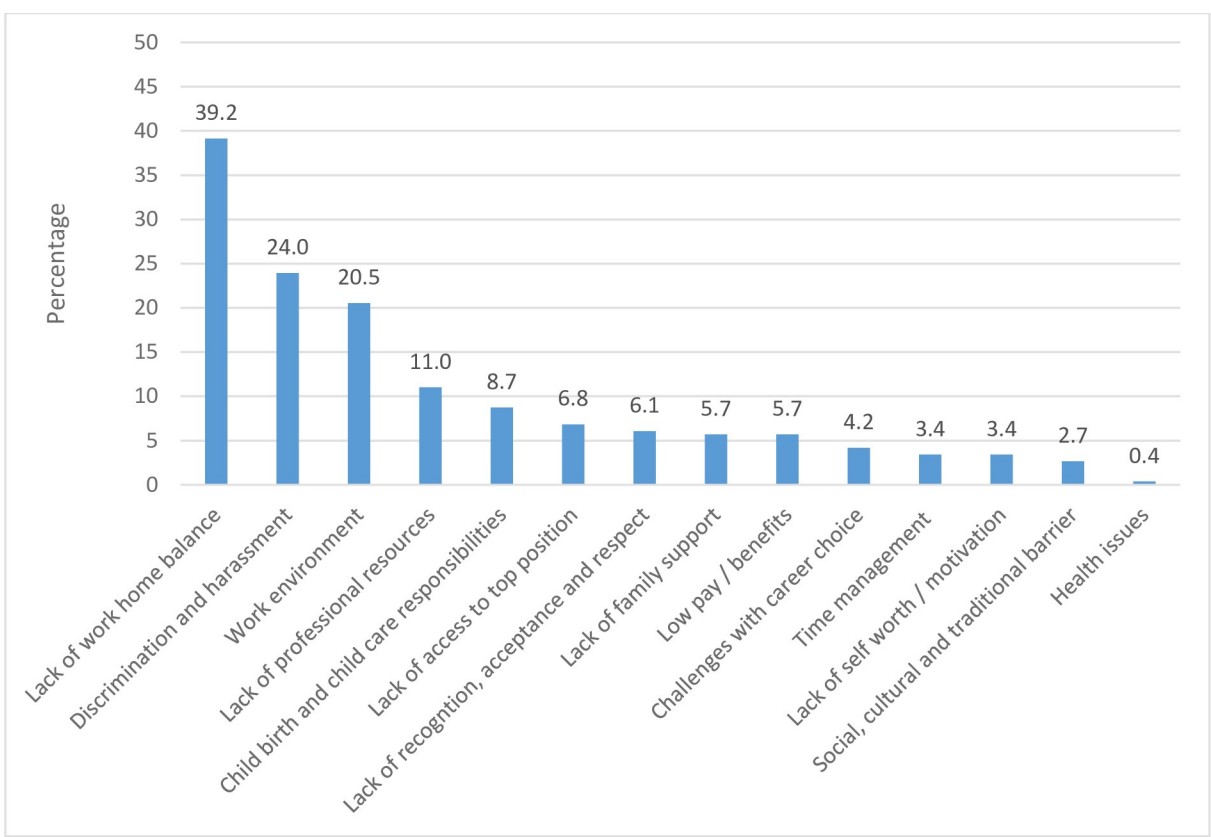

**N=263, Responses are not mutually exclusive**

**Fig 1. Challenges faced by female scientists.**

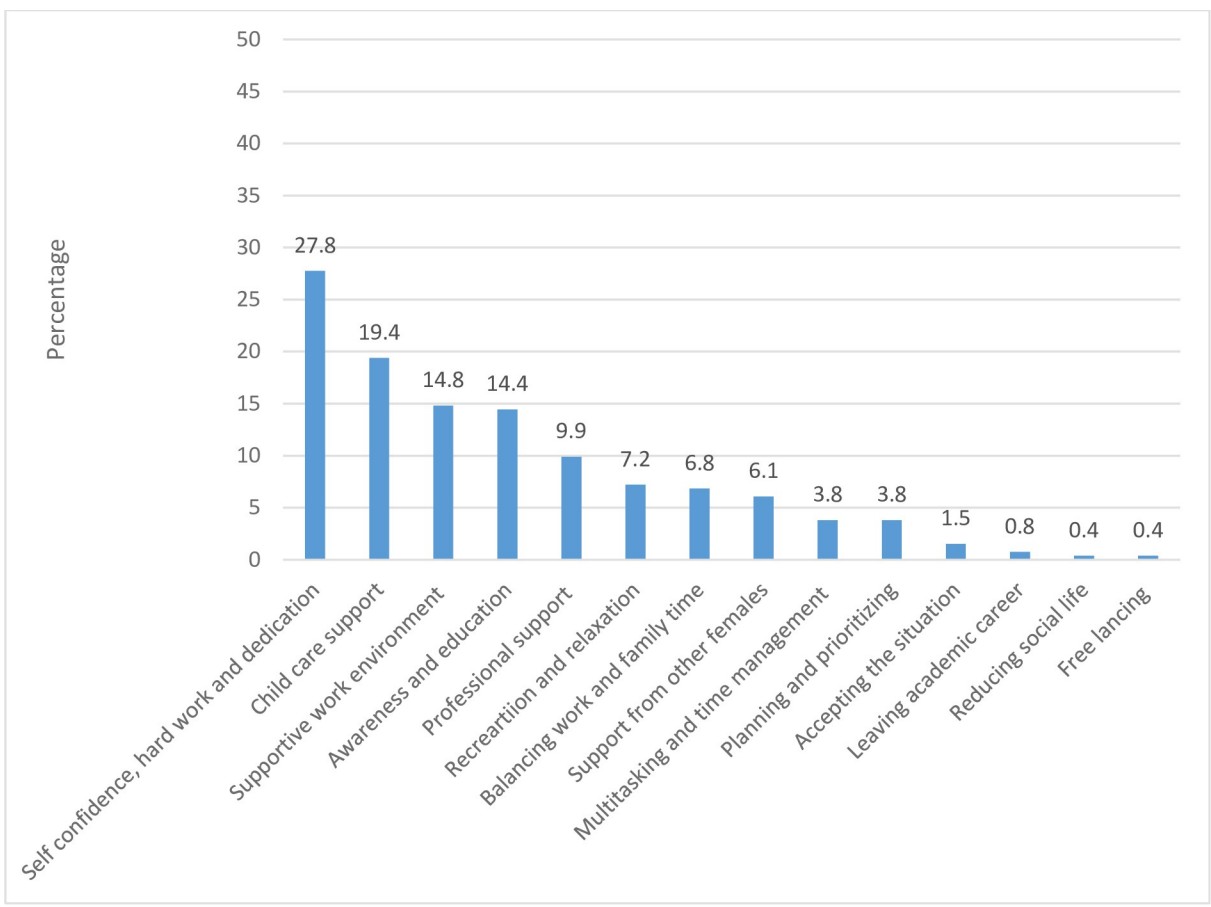

**N=263, Responses are not mutually exclusive**

Fig 2. Coping strategies used by female scientists.

and dedication. Highly career motivated women often are persistent, independent and have high self-esteem [13]. Our study population's coping strategies are in-line with that of successful women entrepreneurs' traits which include self-confidence, honesty, reliability, self-efficacy and strong orientation towards achievement [14]. Some of the other coping strategies operating at an individual level as reported by female scientists in our study include engaging self in recreation activities, relaxation, hobbies and exercise. Similar to our study participants, female domestic workers often find relaxing, being calm, sleeping, talking to someone and doing something they enjoy to be good coping skills [15]. Successful women are often driven by intrinsic motivators and not extrinsic rewards at the workplace [16].

While this is true in case of many successful female scientists, it does not undermine the need to address challenges related to those at workplace and family level.

Many female scientists report using self-motivation to cope with expected challenges. Self-motivation and hard work will help female scientist overcome challenges like maintaining work life balance, taking childcare responsibilities and time management issues. She may even develop resilience to survive in work environments that are not gender sensitive. Available evidence suggests that regardless of educational status, women are more likely to face workplace situations, which mould them to exhibit resilience [17]. Organisational factors such as

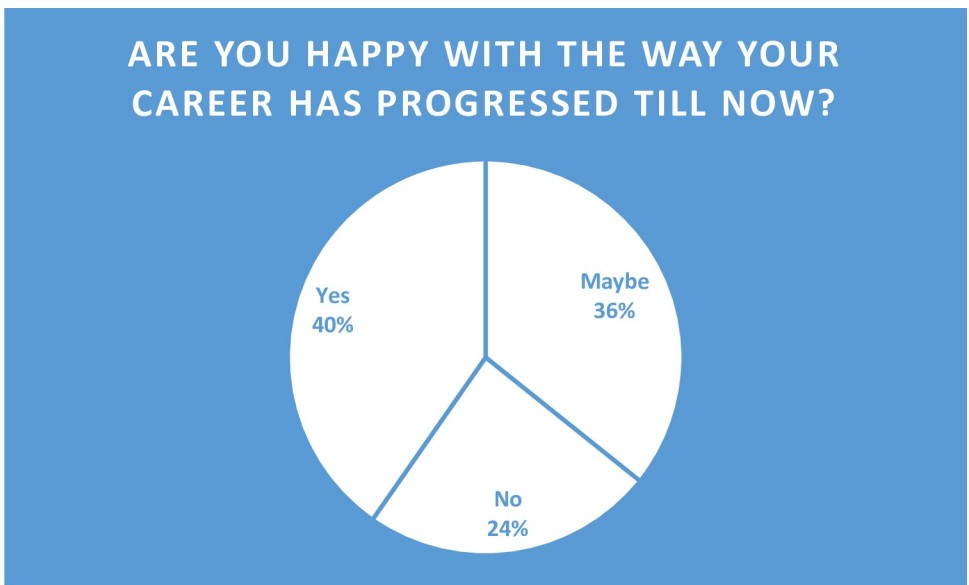

**Fig 3. Happiness with career among female scientists.**

reflective learning, coaching and career counselling and individual factors such as self-esteem, self-efficacy and career adaptability can improve resilience [18].

The second most commonly employed coping strategy was support for childcare. Childcare support could be provided informally by family members or formally by organized childcare services. Pregnancy, breastfeeding and socially driven caretaking of children put women in a disadvantaged position when she has to balance work and family life [19]. Child-care centres are often not found near the workplace, and are costly with long waiting lists. These lead women to work part-time, change residence or change employment. All of these factors might affect her career [20]. Paid maternal and child-care leave allows nurturing the child and mother can resume work with less guilt. However, such leave is variable by region and occupation [21]. Even in developed countries, women are more likely to work before birth (66%) than after birth period (46–49%). This is due to child-care reasons. Even among employed women, taking leave after birth is more frequent due to child-care need [22]. Availability of child-care facilities assists mothers in achieving adequate work-life balance and helps in early childhood development [23]. Strengthening both formal and informal childcare support system will help female scientists to cope with the perceived challenge of child-care responsibilities.

Support at the workplace was the third most commonly employed coping strategy by female scientists in our study. This factor addresses most of the challenges faced by female scientists. The International Labour Organisation (ILO) prohibits discrimination on the grounds of maternity and family responsibilities, as well as sexual harassment [24]. It also advocates for equal remuneration between men and women for work of equal value [25]. The ILO recognises the need to reconcile work with family responsibilities such as childcare, commuting distances for work, part-time work and working hours [26]. These conventions enable women to balance work and life and to avoid unfair treatment in jobs due to their domestic responsibilities. In spite of progress in societal view of gender norms and educational status of women, most organisations have not yet completely grasped the human resource changes and still believe in firm norms [27]. Certain workplaces have flexible working conditions such as working from home and flexitime working which allows women to address work and family issues

[28]. However, even in female majority workplaces, including those in our study inadequate working conditions and low payment are still problems for female scientists [29]. It is imperative that organizations pay attention to making workplaces conducive for female scientists.

The median age of our study population is 35 and 66.2% of them work in junior and middle level jobs. This could be due to the "leaky pipeline" phenomenon where the proportion of women keeps reducing at each level of career [30]. Forty three percent of the study population did not have a child. The exhaustion and stress around child bearing, parenting, and unsupportive work environments could have made female researchers to remain childfree [31]. Despite International Labour standards of 14 weeks of paid maternity leave, breastfeeding breaks and workplace creche for preschool children, in our study we observed that most female scientists take a career break for child bearing and rearing responsibilities [32, 33]. This could be due to the fact that labour standards are not adequately followed in STEM workplaces and hostile work environments.

Mentoring is a character strength and it is often absent in academia due to fierce competition and lack of a mentoring culture [34]. However, in our study we observed that two thirds of female scientists have mentors. This implies that mentoring plays a significant role in female scientists' careers. Most of our study population experienced work related stress. This could be due to women having to balance work and family, high expectations, pay gaps, gap in funding opportunities and harassment at the workplace.

Female scientists, in general have fewer publications compared to their male counterparts [35]. The results of our study depict that female scientists had a median of 8 publications. However, this may not point towards gender bias in the number of publications as the interview tool used in our study was electronic and study participants were not selected randomly, which therefore could have resulted in only junior researchers participating in the study, as reflected in the median age of the study population.

Presentation and participation in scientific conferences provides females with opportunities for networking, recognition from colleagues and future careers prospects. In general, women's representation in conferences is low as compared to men [36]. In our study, we observe that on an average two conferences are attended by the study population. Family responsibilities, difficulty in obtaining sponsorship for conference expenses, well advanced planning for conferences, which will require intrusion into family commitments, institutional academic leave policies and support from senior colleagues are factors, which could play a role in female scientists attending conference.

Most of the study population reported work-related stress. This could be explained by the glass ceiling phenomenon [37]. Gender bias faced during appointments, promotion and career improvement opportunities as well as harassment, biased evaluation by peers and students along with familial responsibilities could cause increased stress among the study population. Financial support from family, household management, safe childcare facilities, flexible work timings to accommodate family duties and support from fellow female researchers can help female scientists achieve work life balance [38]. In our study, nearly half of the study population did not achieve work life balance. This could be due to lack of a female friendly atmosphere, lack of childcare facilities at the workplace, long working hours and poor support from family. In most societies, women, by default have to assume caregiver's role for circumstances such as child-care, elderly-care and during any sickness in the family [39]. This role could affect the female scientists' work performance.

The other commonly cited challenged include discrimination, harassment, lack of professional resources, lack of access to leadership, low pay, career choice challenges, time management and lack of motivation. These issues could be addressed by workplace policies, organizational behaviour change and adherence to the ILO gender equality and

mainstreaming policies [40]. In our study, female scientists often use attributes such as self-confidence, hard work and dedication to ameliorate their work life imbalance. This denotes that the study population often depends on themselves to face work related challenges. Fatoki and Kobiowu observed that support from partners is an important coping strategy for female academics [41]. Similarly, in our study, childcare support and work related support was cited as common coping strategies.

Sexual harassment was reported by 24% of the study population. In workplaces strong policies against sexual harassment, gender balance among staff and training to deal with sexual harassment, strict punishments for perpetrators and adequate reporting can reduce the incidence of sexual harassment [42].

In academic settings, both males and females have the same hierarchal needs of physical safety, love, self-esteem and self-actualisation [18]. A supportive work environment can help to bring out the best in female employees and encourage optimal performance.

### Study limitations

The findings of our study need to be interpreted considering the following limitations. Firstly, the study sample consisted of female scientists who were invited to participate in the study based on personal contacts of the investigators and a snow balling process. This makes the sample selection non-random and limits the external validity of the findings. However, the findings give us some pointers towards the major challenges experienced by female scientists and the coping strategies adopted by them. Secondly, though the sample has representation from all the different WHO regions, the distribution was not proportionally allocated based on population size, resulting in greater representation from some regions and under representation from others. Therefore, the findings of our study may not apply to any specific region and should be viewed as only indicative.

## Conclusion

In conclusion, there is high work related stress, poor work-life balance and sexual harassment experienced by the female scientists in our study. Intrinsic factors like self-motivation, confidence and dedication, as well as extrinsic institutional factors like flexible working time, female friendly management policies, fair appraisal and mentorship appear to improve the stress, work-life balance and therefore productivity of female scientists.

## Supporting information

**S1 Table.**
(XLS)

## Acknowledgments

The authors would like to express their special gratitude to Ms. Emmanuela Oppong for her help and assistance in drawing the graphs.

## Author Contributions

**Conceptualization:** Farah Naaz Fathima, Fathiah Zakham.

**Data curation:** Yi-Chun Yen.

**Formal analysis:** Farah Naaz Fathima, Phyllis Awor, Nancy Angeline Gnanaselvam, Fathiah Zakham.

**Investigation:** Farah Naaz Fathima, Phyllis Awor, Nancy Angeline Gnanaselvam, Fathiah Zakham.

**Methodology:** Farah Naaz Fathima, Nancy Angeline Gnanaselvam.

**Project administration:** Fathiah Zakham.

**Software:** Phyllis Awor, Nancy Angeline Gnanaselvam.

**Supervision:** Fathiah Zakham.

**Validation:** Phyllis Awor, Yi-Chun Yen, Nancy Angeline Gnanaselvam, Fathiah Zakham.

**Visualization:** Phyllis Awor.

**Writing – original draft:** Farah Naaz Fathima, Nancy Angeline Gnanaselvam, Fathiah Zakham.

**Writing – review & editing:** Phyllis Awor, Yi-Chun Yen.

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
