## [Decision Letter · Decision Letter 0]

24 Feb 2020

PONE-D-19-33911

Challenges and coping strategies faced by female scientists - A multicentric cross sectional study

PLOS ONE

Dear Dr Zakham,

Thank you for submitting your manuscript to PLOS ONE. After careful consideration, we feel that it has merit but does not fully meet PLOS ONE’s publication criteria as it currently stands. Therefore, we invite you to submit a revised version of the manuscript that addresses the points raised during the review process.

While Reviewer 1 is satisfied with your contribution, Reviewer 2 raises concerns that you should consider in your revision. Note that Reviewer 2 has provided an attachment with specific comments.  Please let me know if this attachment is not available to you.  I add to these reviews my personal experience in soliciting reviewers for your submission, which is consistent with one of Reviewer 2's comments.  There is a substantial research base on this issue, which is barely acknowledged in the introduction.  Some candidate reviewers commented that the absence of a serious literature review meant that the manuscript did not merit the time to provide a critique.  This is surely not the reaction you seek in a publication.  PLOS-One is primarily data and methodologically oriented.  As such, a review of theory in the area is generally downplayed.  However, it is difficult to see the contribution without some reference to the existing work in this area.  Please conduct the equivalent of a "Google Scholar" search to identify key empirical findings in this area, to foreground your contribution. 

I view the necessary modifications as substantial, and have therefore provided a substantial time window for response.  We would appreciate receiving your revised manuscript by April, 30, 2020. To enhance the reproducibility of your results, we recommend that if applicable you deposit your laboratory protocols in protocols.io, where a protocol can be assigned its own identifier (DOI) such that it can be cited independently in the future. For instructions see: http://journals.plos.org/plosone/s/submission-guidelines#loc-laboratory-protocols

We look forward to receiving your revised manuscript.

Kind regards,

Valerie L. Shalin, Ph.D

Academic Editor

PLOS ONE

Journal Requirements:

2. Please provide additional details regarding participant consent. In the ethics statement in the Methods and online submission information, please ensure that you have specified whether you obtained informed consent from participants.

3. Please include additional information regarding the survey used in the study and ensure that you have provided sufficient details that others could replicate the analyses. For instance, if you developed a questionnaire as part of this study and it is not under a copyright more restrictive than CC-BY, please include a copy, in both the original language and English, as Supporting Information.

4. We noticed you have some minor occurrence(s) of overlapping text with the following previous publication(s), which needs to be addressed:

https://doi.org/10.1053/j.gastro.2013.06.024

https://doi.org/10.1073/pnas.1211286109

https://doi.org/10.1371/journal.pone.0216775

In your revision ensure you cite all your sources (including your own works), and quote or rephrase any duplicated text outside the Methods section. Further consideration is dependent on these concerns being addressed.

6. Please amend either the abstract on the online submission form (via Edit Submission) or the abstract in the manuscript so that they are identical.

Reviewers' comments:

Reviewer's Responses to Questions

**Comments to the Author**

1. Is the manuscript technically sound, and do the data support the conclusions?

Reviewer #1: Yes

Reviewer #2: Yes

2. Has the statistical analysis been performed appropriately and rigorously? 

Reviewer #1: Yes

Reviewer #2: I Don't Know

3. Have the authors made all data underlying the findings in their manuscript fully available?

Reviewer #1: Yes

Reviewer #2: Yes

4. Is the manuscript presented in an intelligible fashion and written in standard English?

Reviewer #1: Yes

Reviewer #2: No

5. Review Comments to the Author

Reviewer #1: This is a potentially important contribution on the topic that requires research and attention of the scientific community. The study is well designed and conducted. However, the quality of reporting can be strengthened further by using the STROBE checklist of items that should be included in reports of observational studies. I would like to encourage the authors to check their manuscript against the STROBE checklist and report the most relevant items in more detail. Additionally, the study would benefit from a more detailed elaboration of the survey questionnaire development. Finally, this recently published article might be of relevance: https://journals.plos.org/plosone/article?id=10.1371/journal.pone.0225763

Reviewer #2: See attached report. You will see that I'm recommending downplaying your quantitative results and instead using them as the basis for an assessment of how different coping strategies do or don't have a prospect of dealing with specific challenges.

6. PLOS authors have the option to publish the peer review history of their article (what does this mean?). If published, this will include your full peer review and any attached files.

Reviewer #1: Yes: Pavel Ovseiko

Reviewer #2: Yes: Brian Martin

---

## [Author Response · Author response to Decision Letter 0]

1 May 2020

Responses to the editor

Comment 1:

While Reviewer 1 is satisfied with your contribution, Reviewer 2 raises concerns that you should consider in your revision. Note that Reviewer 2 has provided an attachment with specific comments. Please let me know if this attachment is not available to you. I add to these reviews my personal experience in soliciting reviewers for your submission, which is consistent with one of Reviewer 2's comments. There is a substantial research base on this issue, which is barely acknowledged in the introduction. Some candidate reviewers commented that the absence of a serious literature review meant that the manuscript did not merit the time to provide a critique. This is surely not the reaction you seek in a publication. PLOS-One is primarily data and methodologically oriented. As such, a review of theory in the area is generally downplayed. However, it is difficult to see the contribution without some reference to the existing work in this area. Please conduct the equivalent of a "Google Scholar" search to identify key empirical findings in this area, to foreground your contribution. 

Response to comment 1:

We thank the editor and the reviewers for their valuable comments. We confirm that have received the attachment of the specific comments of reviewer 2 and we have responded to all his comments.

 As recommended, we paid an attention to a serious literature review and conducted the equivalent of a "Google Scholar" search to identify key empirical findings in this area, to reinforce our contribution, especially in the discussion part. Further, more reference are included.

Comment 2:

I view the necessary modifications as substantial, and have therefore provided a substantial time window for response. We would appreciate receiving your revised manuscript by April, 30, 2020. Response to comment 2:

We agree with the editor about the requested modifications and thank her for the time she provided to revise the manuscript. We have relocated the manuscript in the mentioned folder and we included the corrected manuscript with and without the track changes. We declare that we do not have any change in our previous financial disclosure.

Comment 3:

To enhance the reproducibility of your results, we recommend that if applicable you deposit your laboratory protocols in protocols.io, where a protocol can be assigned its own identifier (DOI) such that it can be cited independently in the future. For instructions see: http://journals.plos.org/plosone/s/submission-guidelines#loc-laboratory-protocols. 

Response to comment 3:

We thank the editor for this comment, but we did not use any laboratory protocols in this study.

Comment 4:

A rebuttal letter that responds to each point raised by the academic editor and reviewer(s). This letter should be uploaded as separate file and labeled 'Response to Reviewers'.

A marked-up copy of your manuscript that highlights changes made to the original version. This file should be uploaded as separate file and labeled 'Revised Manuscript with Track Changes'.

An unmarked version of your revised paper without tracked changes. This file should be uploaded as separate file and labeled 'Manuscript'.

Response to comment 4:

All the requested items are submitted along with revised version of the manuscript. We included a letter to respond to all the points raised by the editor and reviewers. A marked and unmarked versions of the manuscript are also uploaded to the online system of the journal.

 Comment 5:

Response to comment 5:

Well noticed and we also ensured that the revised manuscript is meeting the requirements of the journal. 

1. We ensured that the manuscript meets PLOS ONE's style requirements and we followed the templates of the journal.

2. We obtained a consent from each participant and we mentioned that in the ethics statement.

3. We have developed a questionnaire and all the questions are included in the manuscript.

4. We revised the manuscript to avoid any overlapping with other manuscripts and cited all the resources that we used and all instruction are well respected. 

5. Data Availability statement: the minimal data set underlying the results described in our manuscript are included in the manuscript and uploaded as excel supplementary file and data are fully available. 

6. We ensured that both the abstracts on the online submission form and the manuscript are identical.

7. As recommended, the figure files were uploaded to the Preflight Analysis and Conversion Engine (PACE) digital diagnostic tool, https://pacev2.apexcovantage.com/ to meet the requirements of Plos One.

Response to Reviewers:

Reviewer 1: 

Comment 1:

This is a potentially important contribution on the topic that requires research and attention of the scientific community. The study is well designed and conducted. However, the quality of reporting can be strengthened further by using the STROBE checklist of items that should be included in reports of observational studies. I would like to encourage the authors to check their manuscript against the STROBE checklist and report the most relevant items in more detail. Additionally, the study would benefit from a more detailed elaboration of the survey questionnaire development. Finally, this recently published article might be of relevance: https://journals.plos.org/plosone/article?id=10.1371/journal.pone.0225763

Response to comment 1:

We thank the reviewer for the comment and the attached manuscript. We have used the STROBE checklist items for observational studies wherever applicable to our study.

Reviewer 2:

General comment:

 See attached report. You will see that I'm recommending downplaying your quantitative results and instead using them as the basis for an assessment of how different coping strategies do or don't have a prospect of dealing with specific challenges.

Response to the general comment:

The authors agree with reviewer about the value of using the quantitative data for the assessment of coping strategies in dealing with the specific challenges and the authors considered this remark in the revised manuscript.

Comment 1: Strengths of the paper

The feminist movement has drawn attention to the disadvantages faced by women in a wide range of domains, of which science is an important one. Scientific careers seem to have been especially difficult for women, even though science might not seem to be so obviously a masculine domain. There have been many studies of the challenges faced by women in science.

This paper offers several valuable new angles to this issue. It analyses data from female scientists from around the world, especially in south and Southeast Asia. This geographical area warrants attention. The workplace factors raised provide valuable insight into the careers and concerns of female scientists. Especially useful are the findings concerning coping strategies, something given little attention in most studies.

After a brief setting of the stage, the study and its methods are clearly described. The findings are presented with adequate detail. 

Response to comment 1:

The authors would like to thank referee for this comment. In fact as women scientists, we feel that women in science are still underrepresented, even in the developed world and we wanted to learn from other female scientists around the world about the challenges that they face and the coping strategies that they use to deal with.

Comment 2: Suggestions for improvement:

Comment 2.1:

Given that the sample of female scientists is not random, but rather a convenience sample, it would be better to present the findings as indicative of major concerns in the cohort. Detailed figures and percentages are of less significance, and serve mainly to indicate the likely findings from a random sample. Furthermore, because the sample is worldwide, though with heavier representation from some areas, the findings cannot readily be said to apply to any specific area. Similarly, the findings about coping strategies are only indicative. Nevertheless, they reveal self-understandings by female scientists.

To make the paper stronger, I think it would be useful for the authors to make their own assessments of how the coping strategies relate to the challenges. The most commonly cited coping strategies could be assessed in terms of whether they have the potential to address the challenges effectively. For example, one of the common challenges is workplace harassment. The authors could examine each coping strategy to determine whether it has a reasonable potential to address harassment. Hard work and dedication may not help a lot, and indeed harassment is often designed to undermine someone’s performance. Child care support seems unrelated to addressing harassment, whereas a supportive work environment would make a big difference.

This little example then raises the question of whether a supportive work environment constitutes a strategy. Choosing a job where the environment is supportive could reduce the risk and impact of harassment, and trying to foster a more supportive environment might do the same.

The point here is not this example but the value of using the findings not as data giving accurate figures but as indicators of salient issues, specifically challenges and coping strategies. Assessing the main coping strategies in relation to the main challenges would make the study much more worthwhile. 

Response to comment 2.1:

As recommended, the authors used their assessment and experience in fieldwork on relating the coping strategies to the challenges. This point is discussed carefully for each coping strategy and updated studies references were used to emphasize the role supportive work environment in undermining the overwhelming challenges that female scientists face at familial and professional level.

Comment 2.2:

The authors might also use their collective experience and understanding to suggest strategies that would be more effective than those commonly reported.

In summary, the figures can be downplayed, the most common challenges and coping strategies highlighted, and connections between the challenges and strategies outlined.

Response to comment 2.2:

The authors have considered this point in the modified version and extensively discussed the different effective strategies that they have experienced to enrich the content of the manuscript. 

Comment 3 Lesser points: English expression should be polished.

Response to the comment 3 of lesser point: The English language has been revised by all the authors and an English native speaker.

---

## [Decision Letter · Decision Letter 1]

21 Jul 2020

PONE-D-19-33911R1

Challenges and coping strategies faced by female scientists - A multicentric cross sectional study

PLOS ONE

Dear Dr. Zakham,

Thank you for submitting your manuscript to PLOS ONE. After careful consideration, we feel that it has merit but does not fully meet PLOS ONE’s publication criteria as it currently stands. Therefore, we invite you to submit a revised version of the manuscript that addresses the points raised during the review process.

Thank you for your careful attention to reviewer comments.  The manuscript is much improved.  During this final iteration of our exchange, I focus on expository considerations.  Most important is the addition of subheadings on in your methods, results and discussion.  (An aside: Please title the Method section "Methods" rather than "Methodology" which refers to the study of method.) Subheadings help the reader locate information and summarize your main points.  The titles of your figures and tables provide a guideline for these, although the discussion can call out "limitations".  Please change the nationality  indication to country names rather than adjectives.  Canadians believe they are Americans!  If you change the language from adjective (Brazilian) to countries (Brazil) you can refer to the United States explicitly. Finally, please have a native English speaker review the manuscript.  Although the language is generally quite good, the occasional missed article (line 179 should be "A majority") detracts from readability.  These are minor revisions and will not require another round of reviewer comments. 

We look forward to receiving your revised manuscript.

Kind regards,

Valerie L. Shalin, Ph.D

Academic Editor

PLOS ONE

Reviewers' comments:

Reviewer's Responses to Questions

**Comments to the Author**

1. If the authors have adequately addressed your comments raised in a previous round of review and you feel that this manuscript is now acceptable for publication, you may indicate that here to bypass the “Comments to the Author” section, enter your conflict of interest statement in the “Confidential to Editor” section, and submit your "Accept" recommendation.

Reviewer #2: All comments have been addressed

2. Is the manuscript technically sound, and do the data support the conclusions?

Reviewer #2: Yes

3. Has the statistical analysis been performed appropriately and rigorously? 

Reviewer #2: I Don't Know

4. Have the authors made all data underlying the findings in their manuscript fully available?

Reviewer #2: Yes

5. Is the manuscript presented in an intelligible fashion and written in standard English?

Reviewer #2: Yes

6. Review Comments to the Author

Reviewer #2: Please see attached reviewer report. Please note my preference not to make a formal recommendation, and rely only on my report.

7. PLOS authors have the option to publish the peer review history of their article (what does this mean?). If published, this will include your full peer review and any attached files.

Reviewer #2: **Yes: **Brian Martin

---

## [Author Response · Author response to Decision Letter 1]

31 Jul 2020

Responses to the editor

Comment 1:

 Thank you for submitting your manuscript to PLOS ONE. After careful consideration, we feel that it has merit but does not fully meet PLOS ONE’s publication criteria as it currently stands. Therefore, we invite you to submit a revised version of the manuscript that addresses the points raised during the review process. Thank you for your careful attention to reviewer comments. The manuscript is much improved. During this final iteration of our exchange, I focus on expository considerations. Most important is the addition of subheadings on your methods, results and discussion. (An aside: Please title the Method section "Methods" rather than "Methodology" which refers to the study of method.) Subheadings help the reader locate information and summarize your main points. The titles of your figures and tables provide a guideline for these, although the discussion can call out "limitations". 

Response to comment 1: 

We thank the editor and the reviewers for the efforts that they made to improve the quality of the paper. As recommended, we have added more subheadings in all requested sections and we changed "Methodology" to "Methods".

Comment 2:

Please change the nationality indication to country names rather than adjectives. Canadians believe they are Americans! If you change the language from adjective (Brazilian) to countries (Brazil) you can refer to the United States explicitly. 

Response to comment 2: 

The nationalities were replaced by the country names and we also double checked and included Algeria in Africa, since it has not been considered in Eastern Mediterranean Region like other neighboring countries like Tunisia and Morocco. Further, Filipino is removed and included with Philippines.

Comment 3:

Finally, please have a native English speaker review the manuscript. Although the language is generally quite good, the occasional missed article (line 179 should be "A majority") detracts from readability. These are minor revisions and will not require another round of reviewer comments. 

Response to comment 3: 

We have checked the manuscript with an English native speaker and line 179 is corrected.

Comment 4:

Response to comment 4: 

All the requested items are submitted along with revised version of the manuscript. We included a letter to respond to all the points raised by the editor and reviewer. A marked-up and unmarked versions of the manuscript are also uploaded to the online system of the journal.

We declare that we do not have any change in our previous financial disclosure. We also confirm that we did not use any laboratory protocols in this study.

Comment 5:

Response to comment 5:

We confirm that have received the attachment of the minor comments of one reviewer and we have responded to all his comments.

Comment 6:

Response to comment 6:

The figure files were uploaded to the Preflight Analysis and Conversion Engine (PACE) digital diagnostic tool, https://pacev2.apexcovantage.com/ to meet the requirements of Plos One.

………………………………………………………………………………………………………………………………………………

Response to Reviewer:

Reviewer: 

Comment 1: Improvements in the revised paper

The paper provides a window into the challenges facing female scientists from many parts of the world, especially southeast Asia, and their coping strategies. This is a valuable insight into the current situation for women in science. The revised version of the paper addresses my suggestions for improvement. It uses descriptive statistics from the survey to highlight frequently reported challenges and coping strategies.

Response to comment 1:

We glad that the reviewer is satisfied with the corrections that we made and thank him for the additional minor comments.

Comment 2: Minor points:

• Line 58, “paid” should be “pay”.

• Line 250, “Our results indicate that if the female scientist is self-motivated, she will be driven to cope with the expected challenges that comes her way.” A better way to express this point might be, “Many female scientists report using self-motivation to cope with expected challenges.”

• Line 351, [411] should be [41].

Response to comment 2:

All the requested modifications in lines 58, 250 and 351have been made, but line numbers have been shifted in the modified version.

---

## [Editor Report · Decision Letter 2]

21 Aug 2020

Challenges and coping strategies faced by female scientists - A multicentric cross sectional study

PONE-D-19-33911R2

Dear Dr. Zakham,

We’re pleased to inform you that your manuscript has been judged scientifically suitable for publication and will be formally accepted for publication once it meets all outstanding technical requirements.

Kind regards,

Valerie L. Shalin, Ph.D

Academic Editor

PLOS ONE
---

## [Editor Report · Acceptance letter]

11 Sep 2020

PONE-D-19-33911R2 

Challenges and coping strategies faced by female scientists - A multicentric cross sectional study 

Dear Dr. Zakham:

I'm pleased to inform you that your manuscript has been deemed suitable for publication in PLOS ONE. Congratulations! Your manuscript is now with our production department. 

Kind regards, 

on behalf of

Dr. Valerie L. Shalin 

Academic Editor

PLOS ONE